# Explaining the 'Outliers' Track in Black Hole X-ray Binaries with a BZ-Jet and Inner-Disk Coupling

Ning Chang [1,2], Xiang Liu [1,3,4,*], Fu-Guo Xie [5], Lang Cui [1,3,4] and Hao Shan [1]

1   Xinjiang Astronomical Observatory, Chinese Academy of Sciences, 150 Science 1-Street,
    Urumqi 830011, China; changning@xao.ac.cn (N.C.); cuilang@xao.ac.cn (L.C.); shanhao@xao.ac.cn (H.S.)
2   School of Astronomy and Space Science, University of Chinese Academy of Sciences, Beijing 100049, China
3   Key Laboratory of Radio Astronomy, Chinese Academy of Sciences, Nanjing 210008, China
4   Xinjiang Key Laboratory of Radio Astrophysics, Urumqi 830011, China
5   Key Laboratory for Research in Galaxies and Cosmology, Shanghai Astronomical Observatory,
    Chinese Academy of Sciences, 80 Nandan Road, Shanghai 200030, China; fgxie@shao.ac.cn
*   Correspondence: liux@xao.ac.cn

**Abstract:** In this paper, we investigate the black hole (BH) spin contribution to jet power, especially for the magnetic arrested disk (MAD), where only inner accretion disk luminosity is closely coupled with the spin-jet power, and try to explain the 'outliers' track of the radio $L_R$ to X-ray luminosity $L_X$ in two black hole X-ray binaries (BHXBs). Our results suggest that the BZ-jet and the inner-disk coupling could account for the 'outliers' track of the radio/X-ray correlation in two BHXBs, H1743-322 and MAXI J1348-630. Although the accretion disk of H1743-322 in the outburst could be in the MAD state, there is a lower probability that MAXI J1348-630 is in the MAD state due to its low jet production efficiency. The difference in the inner-disk bolometric luminosity ratio of the two sources implies that these two BHXBs are in different inner-disk accretion states. We further investigate the phase-changing regime of MAXI J1348-630 and find that there is a phase transition around $L_X/L_{Edd} \sim 10^{-3}$. The assumption of sub-MAD is discussed as well.

**Keywords:** accretion disk; black hole physics; X-ray binaries; radio jets

## 1. Introduction

Black hole (BH) X-ray binaries (BHXBs) have been found in our own galaxy as well as nearby galaxies. About twenty have been dynamically confirmed (https://www.astro.puc.cl/BlackCAT/transients.php, accessed on 22 May 2022) [1] to be hosted by BHs with a mass of $M_{BH} \sim$ 3–20$M_\odot$, typically $\sim 10M_\odot$, which are accreting gas from a companion star and emitting transient X-ray emission; compact radio jets are often observed as well. Strong correlations between radio luminosity ($L_R$) and X-ray luminosity ($L_X$) have been found, and these can be well described in the power-law form ($L_R \propto L_X^\mu$).

Certain BHXBs follow the 'standard' track, with $\mu \sim 0.6$ [2,3]. This correlation is shown in low-luminosity active galactic nuclei as well [4,5]. Other sources follow an 'outliers' track, with $\mu \gtrsim 1$ [6,7], and sometimes a transition phase between the two different tracks is observed [6,8,9]. We call this a 'hybrid' correlation, which consists of three power-law branches, i.e., the index of the correlation varies in different regimes of luminosity. It has been proposed that the radiative inefficient hot accretion flow (RIAF) could be responsible for the 'standard' track, while the inner thin-disk component or luminous hot accretion flow could account for the 'outlier' track [10–12]. In addition, black hole spin with an accreted magnetic flux can contribute to a radio jet [13–15], which may lead to a correlation between the radio and inner accretion disk (hard X-ray luminosity) [16,17] similar to the relativistic jets in AGN or blazars, which may themselves be powered by BH spin [18]. However, these explanations are mostly not well justified by quantitative analysis, especially for the effect of BH spin on the radio/X-ray correlation of BHXBs. Developed from the Blandford–Znajek

(BZ) mechanism [13], the magnetically arrested disk (MAD) (e.g., [19–23]) has attracted much attention in recent years because it is believed to be able to drive more powerful jets than the normal and standard evolution (SANE) of the disk [24] via the extraction of the additional energy from black hole spin (e.g., [14]). The MAD state may be more easily achieved in the hot accretion flow around a spinning BH [15] than previously thought. In this work, we intend to analyze the potential contribution of BH spin to the correlation of radio $L_R$ and X-ray luminosity $L_X$ observed in two BHXBs, H1743-322 and MAXI J1348-630, which have a prominent 'outliers' tracks in their radio/X-ray correlations.

There have been suggestions of correlation between observed radio jet power and the BH spin in BHXBs. Narayan and McClintock (2012) [24] found that jet power scales either as the square of dimensionless BH spin parameter $a_*$ or as the square of the angular velocity of the BH horizon $\Omega_H$ based on data from four BHXBs; they used the peak luminosity at 5 GHz as a proxy for jet power (normalized by BH mass) and estimated $a_*$ via the continuum-fitting method. Steiner et al. (2013) [25] tested and confirmed the above empirical relationship using a fifth source, H1743-322, by measuring its spin, and showed that this relationship is consistent with the BH spin measured by Fe line. They further suggested, using the standard synchrotron bubble model, that the radio luminosity at the light-curve maximum is a good proxy for jet kinetic energy. However, Russell et al. (2013) [26] argued that the peak radio flux differs dramatically depending on the outburst (up to a factor of 1000), whereas the total power required to energize the flare may only differ by a factor of $\lesssim 4$ between outbursts. They claim that if the peak flux is determined by the total energy in the flare and the time over which it is radiated (which can vary considerably between outbursts), then, using a Bayesian fitting routine, they are able to rule out a statistically significant positive correlation between transient jet power measured using these methods and estimates of BH spin (either with the continuum-fitting method or the reflection model), based on a larger sample of twelve BHXBs.

It was predicted by Blandford and Znajek (1977) [13] that jet power scales with the square of the magnetic flux ($\Phi$) threading on the horizon of a BH. This was revised in the form of $P_{BZ} \approx (\kappa/4\pi c)\Omega_H^2\Phi^2$, $\kappa \approx 0.05$, $\Omega_H \approx a_*c/2R_H$, where $R_H$ is the radius of the horizon of the BH [27–29] and $\Phi$ should be related to the magnetic field $B$ and accretion rate $\dot{M}$. Thus, there is no simple relation between jet power and the BH spin parameter $a_*$ for different BHXBs with various $B$ and $\dot{M}$. Sikora and Begelman (2013) [30] argued that jet power is mostly driven by the magnetic flux, not the spin, in the hard state of BHXBs. It may be the case that $B^2 \propto \dot{m}$, the mass accretion rate in Eddington accretion rate unit (e.g., [31]).

In this work, we intend to interpret the 'outlier' track of BHXBs based on the BZ-jet and inner disk in order to provide an analysis of the possible contribution from the BZ-jet to the $L_R - L_X$ correlations. We focus on two BHXB sources, H1743-322 and MAXI J1348-630 [6,32], both of which have a hybrid radio/X-ray correlation. As regards H1743-322, the first outburst occurred in 2003, and the following flares lasted for several years. Recently, Williams et al. (2020) [33] reported the 2018 outburst of H1743-322, which seemed to be independent from the early burst. The 2018/2019 outburst of MAXI J1348-630 was reported by Carotenuto et al. (2021) [32], including one main outburst and seven flares.

We organize this paper as follows. In Section 2, we briefly introduce the BZ-jet in the MAD state, while in Section 3 we present the results for two BHXBs. A short discussion is provided in Section 4 and a brief summary in Section 5.

## 2. BZ-Jet in MAD State

Magnetically dominated hot accretion flows have been developed for a magnetically arrested disk (MAD) [19–22,34], or a magnetically choked accretion flow [23]. When a significant amount of magnetic flux threads the horizon, the magnetic field outside the black hole becomes so strong that it forces the gas to move inward via streams and blobs, as was later confirmed in 3D GRMHD simulations [14,23,35]. The MAD state is special in

that the flux threading the hole is at its maximum saturation value ($\Phi_{\mathrm{MAD}}$) for the given mass accretion rate $\dot{M}$. This saturation flux is approximately [14,35]:

$$
\begin{aligned}
\Phi_{\mathrm{MAD}} &\approx 50\dot{M}_{\mathrm{BH}}^{1/2}R_{\mathrm{g}}c^{1/2} \\
&= 1.5 \times 10^{21}(M_{\mathrm{BH}}/M_\odot)^{3/2}(\dot{M}_{\mathrm{BH}}/\dot{M}_{\mathrm{Edd}})^{1/2}\mathrm{Gauss}\ cm^2,
\end{aligned}
\tag{1}
$$

and with the corresponding field strength $B$ at the horizon of roughly $B \approx \Phi_{\mathrm{MAD}}/2\pi R_{\mathrm{g}}^2$, where $\dot{M}_{\mathrm{Edd}}$ is the Eddington accretion rate and $R_{\mathrm{g}} = GM/c^2$ is the gravitational radius of BH. In the BZ-jet, the jet power $P_{\mathrm{jet}}$ depends on the BH spin parameter $a_*$, the magnetic flux ratio ($\Phi/\Phi_{\mathrm{MAD}}$) threading on BH, and the mass accretion rate $\dot{M}$ [14,28], i.e.,

$$
P_{\mathrm{jet}} \approx 0.65a_*^2(1 + 0.85a_*^2)(\Phi/\Phi_{\mathrm{MAD}})^2\dot{M}_{\mathrm{BH}}c^2.
\tag{2}
$$

This expression works for either low-$\Phi$ hot accretion flow or the MAD state ($\Phi/\Phi_{\mathrm{MAD}} \approx 1$).

It has been suggested that the MAD state can be easily achieved in a hot accretion flow around a spinning BH [15,36]. As shown in Equation (1), the $\Phi_{\mathrm{MAD}}$ depends on the accretion rate, $\dot{M}_{\mathrm{BH}}/\dot{M}_{\mathrm{Edd}}$, which can range from $10^{-5}$ at a low accretion rate to 0.1 at a high accretion state; however, the BH mass is almost the same within a factor of three in BHXBs. Thus, the MAD state can be reached at both a high accretion rate (e.g., in a thin-disk component or a luminous hot accretion flow) and at a low accretion state (a normal hot accretion flow) if a sufficient magnetic flux has been threading the BH. Thus, we adopt Equation (2) to analyze the possible correlation between radio and inner X-ray luminosity assuming the MAD state. We caution that the early phase of the outburst is improbable in the MAD state if there is a long-term quiescent state before the outburst of the BHXB, as the central magnetic field will diffuse and become weaker in the quiescent state. For the quiescent states between the repeat minor flares, it is difficult to determine whether or not these minor flares are in a MAD state.

In the following, we use Equation (2) for the MAD state ($\Phi/\Phi_{\mathrm{MAD}} = 1$) to analyze the possible correlation between radio and X-ray luminosity. In this condition, there are only two remaining free parameters: the spin parameter $a^*$ and the accretion rate $\dot{M}_{\mathrm{BH}}$. Xie and Yuan (2012) [37] presented the fitting formula of radiative efficiency $\varepsilon$ as a function of accretion rate for different electron viscous heating parameters, where the radiative efficiency $\varepsilon$ is defined as

$$
\varepsilon = \frac{L_{\mathrm{bol}}}{\dot{M}_{\mathrm{BH}}c^2},
\tag{3}
$$

where $L_{\mathrm{bol}}$ is the bolometric luminosity. We note that the BZ-jet is formed by extracting the additional spin energy of the BH; thus, the $\dot{M}_{\mathrm{BH}}$ in Equation (2) is the inner mass accretion rate (e.g., around the innermost stable circular orbit (ISCO) to the BH), with $L_{\mathrm{in}} = \varepsilon\dot{M}_{\mathrm{BH}}c^2$ where $L_{\mathrm{in}}$ is the inner-disk luminosity. Because of outflows, the mass accretion rate will decrease, as $\dot{M}_{\mathrm{BH}} \propto r^s$ with $s \sim 0.4$–$0.5$ [38], where $r$ is the disk distance from the central BH. Thus, the inner-disk luminosity is only a fraction of the bolometric luminosity. Here, we assume it as $L_{\mathrm{in}} \approx f_{\mathrm{in}}L_{\mathrm{bol}}$ with $f_{\mathrm{in}} < 1$. This factor of $f_{\mathrm{in}}$ can be estimated from Equation (2) if other parameters are measured or constrained. Together with Equation (2), we then have

$$
P_{\mathrm{jet}} \approx 0.65a_*^2(1 + 0.85a_*^2)\frac{L_{\mathrm{in}}}{\varepsilon},
\tag{4}
$$

which we can use to obtain the jet power $P_{\mathrm{jet}}$ (or the inner-disk luminosity $L_{\mathrm{in}}$) for a comparison with the observed correlation between radio and X-ray luminosity, in which the jet power $P_{\mathrm{jet}}$ is related to the radio luminosity $L_{\mathrm{R}}$ and the inner-disk luminosity $L_{\mathrm{in}}$ (or the bolometric luminosity $L_{\mathrm{bol}}$) is related to the X-ray luminosity $L_{\mathrm{X}}$. The BH spin parameter $a_*$ can be found in the literature, and can be measured by disk continuum fitting, the reflection method, or an Fe line (e.g., [39–41]).

In order to make the comparison between the radio luminosity and jet power, we need to convert the radio luminosity to the theoretical jet power. Heinz and Grimm (2005) [42] estimated the relation of radio luminosity and jet power based on the formula of kinetic jet power calibrated with Cyg X-1 and GRS 1915+105, with a formation akin to (e.g., [16])

$$L_{\rm R} = 6.1 \times 10^{-23} P_{\rm jet}^{17/12} \, {\rm erg \, s}^{-1}, \tag{5}$$

because the radio emission of BHXBs in the hard state is often an optically thick core emission.

## 3. Results

The radio/X-ray planes of the BHXBs H1743-322 and MAXI J1348-630 are shown in Figure 1. The data for H1743-322 and MAXI J1348-630 were obtained from Islam and Zdziarski (2018) [43] and Carotenuto et al. (2021) [11], respectively. We converted the radio luminosity ($L_{\rm R}$) of both sources into 5 GHz luminosity, assuming a flat radio spectral index (a measured spectral index was used for MAXI J1348-630 [11]), and converted the 3–9 keV X-ray luminosity ($L_{\rm X}$) to 1–10 keV, assuming a typical photon index of 1.8. Although both sources show 'hybrid' correlations, there are several differences between these two BHXBs. First, there seems to be no obvious transition in MAXI J1348-630 compared to H1743-322, which has a sharp transition at $L_{\rm X} \sim 4 \times 10^{36}$ erg s$^{-1}$ [6], corresponding to $\sim 3 \times 10^{-3} L_{\rm Edd}$. Moreover, the radio luminosity of MAXI J1348-630 is about one magnitude lower than H1743-322 at low X-ray luminosity.

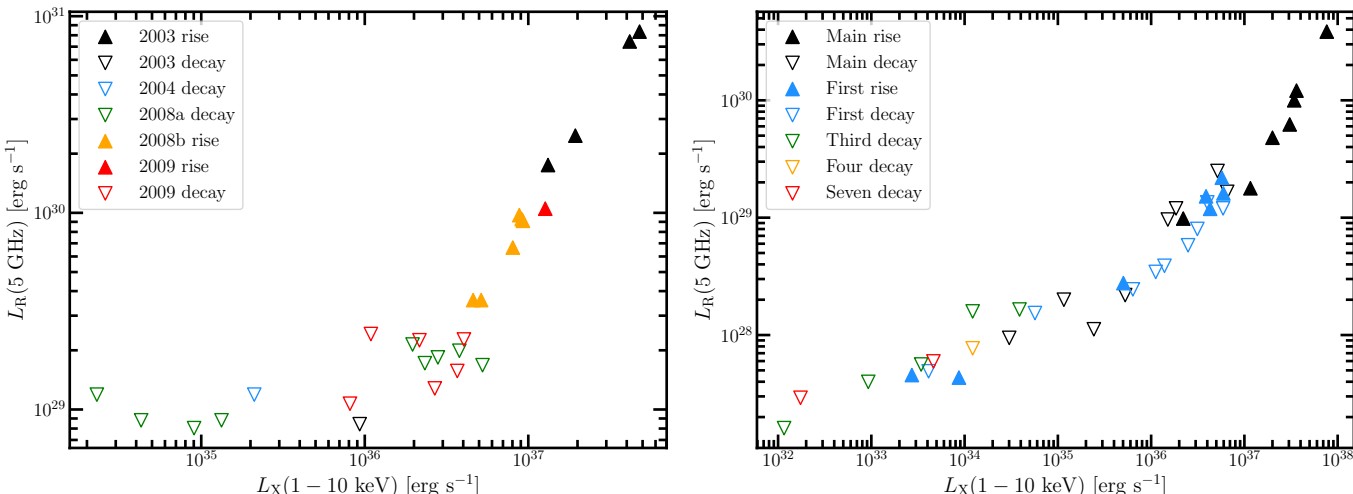

**Figure 1.** The respective radio/X-ray planes of the BHXBs H1743-322 and MAXI J1348-630 are shown in the left panel and right panel. In both panels, different triangles pointing upwards or downwards differentiate the outburst phase, i.e., filled triangles pointing upwards for the rise and open triangles pointing downwards for decay; the colors represent the different outbursts.

In the following, we try to calculate the inner-disk luminosity $L_{\rm in}$ of the 'outlier' track of the two BHXBs using the observed radio luminosity with Equations (4) and (5). We note that there is an unknown variable, $\varepsilon$, in Equation (4) except for the other measurements ($L_{\rm R}$ and $a_*$). The radiative efficiency $\varepsilon$ of a geometrically thin cold accretion disk, such as the Shakura–Sunyaev disk (SSD, [44]), is ~0.1. For a hot accretion flow, the efficiency is systematically lower than the SSD, and the radiative efficiency $\varepsilon$ should be <0.1; however, its value increases as $\dot{M}_{\rm net}$ (the innermost mass accreted onto BH) increases from 0.001 up to a plateau of nearly constant value $\varepsilon \approx 0.08$ in the high accretion regime [37,45]. In order to interpret the 'outlier' track of BHXBs under the BZ-jet in the MAD state, where the accretion rate is high, we use $\varepsilon = 0.08$ in this work.

*3.1. H1743-322*

The BHXB H1743-322 was discovered with the Ariel V and HEAO-1 satellites during a bright outburst in 1977 [46]. In 2003, another bright outburst was first detected with the International Gamma-ray Astrophysics Laboratory (INTEGRAL). The major outburst in 2003 was followed by five minor activity periods between 2004 and 2009. The source showed a typical 'outliers' track in its radio/X-ray correlation [6]. This 'outlier' behavior was recently observed by MeerKAT in the 2018 outburst of H1743-322 [33]. The distance, jet inclination angle, BH mass, and BH spin of H1743-322, respectively, are $8.5 \pm 0.8$ kpc (with its location near the Galactic centre), $75 \pm 3$ deg [47], $\sim 10 M_\odot$ (assumed in [43]), and $0.2 \pm 0.3$ (with disk continuum fitting at a confidence level of 68% [47]).

We first investigate the distribution of the bolometric luminosity ratio $L_{in}/L_{bol}$ as a function of the $L_X/L_{Edd}$; the bolometric luminosity $L_{bol}$ was obtained from [43]. As shown in Figure 2, we found that, statistically, there is no dependence of the ratio $L_{in}/L_{bol}$ on the $L_X/L_{Edd}$. Moreover, there is a slight uptrend at $L_X/L_{Edd} > \sim 10^{-2}$, although with a large scatter. We show the X-ray/bolometric luminosity ratio $L_X/L_{bol}$ as a function of the $L_X/L_{Edd}$, which shows the same trend as the bolometric luminosity ratio, although again with a smaller scatter. We found that $L_{in}/L_{bol}$ varies between $\sim 0.2$ and $\sim 0.8$, with a mean value of $<L_{in}/L_{bol}> = 0.391 \pm 0.185$; in other words, the inner-disk produces $\sim 40$ percent of the luminosity of the whole accretion disk. The $L_X/L_{bol}$ value follows the same pattern as the bolometric luminosity ratio, although with a slightly shallower mean value of $<L_{in}/L_{bol}> = 0.191 \pm 0.081$. Another result based on this plot is that the tendency of $L_{in}/L_{bol}$ to vary with $L_X/L_{Edd}$ is extremely similar to that of $L_X/L_{bol}$. This might suggest that if the BHXB is in the MAD state, the inner-disk brightens at the same pace as the $L_X$ and remains relatively steady.

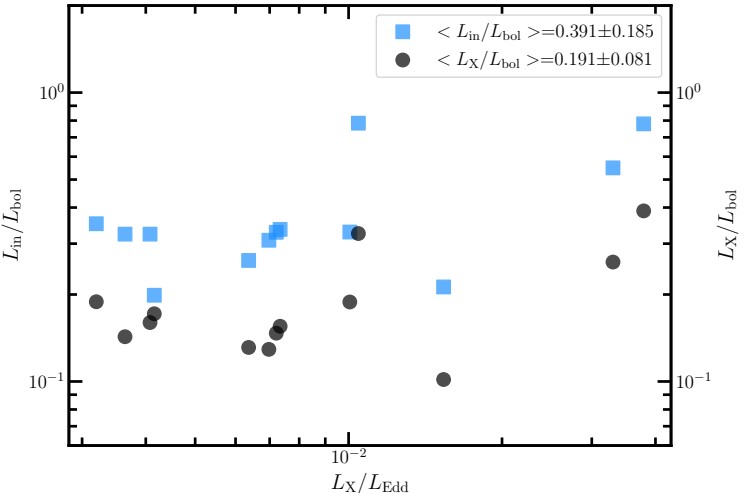

**Figure 2.** Distribution of the inner-disk luminosity to the bolometric luminosity ratio $L_{in}/L_{bol}$ versus 1–10 keV X-ray luminosity (in Eddington units) in the 'outlier' track of H1743-322 (in the right y-axis, the 1–10 keV X-ray luminosity to bolometric luminosity ratio is $L_X/L_{bol}$). As labelled in the plot, the blue squares stand for $L_{in}/L_{bol}$ and the black circles for $L_X/L_{bol}$.

Figure 3 shows the plot of the relationship between $L_R$ and inner-disk luminosity $L_{in}$ for the BZ-jet in the MAD state with a comparison to the radio/X-ray plane for the 'outlier' track, with blue squares for $L_{in}$–$L_R$ and black circles for $L_X$–$L_R$. Several results can be derived directly from this plot. First, the result of the radio/inner-disk correlation shows good consistency with the observational data if we apply the correction $L_X/L_{in} \approx 0.5$ from Figure 2. This suggests that H1743-322 might be in the MAD state during its outburst. We notice from Steiner and McClintock (2012) [47] that while H1743-322 has a mildly spin parameter $a_* \sim 0.2$, it is nonetheless able to produce a strong jet compared to MAXI J1348-630 (with $a_* \sim 0.78$, [48]) if there is a strong magnetic flux (i.e., under the MAD state).

From this, it can be argued that the magnetic flux must play a dominant role in jet power (e.g., [49]), while the BH spin plays an important role in jet power if the BH is under the MAD state, as per Equation (2).

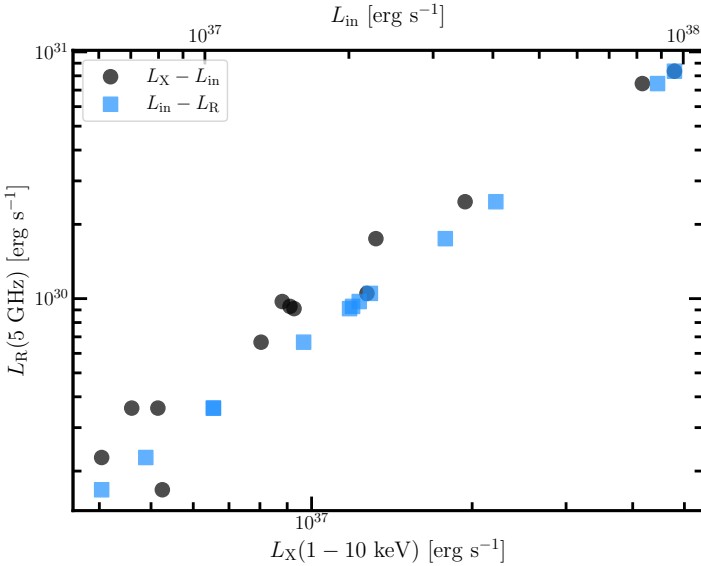

**Figure 3.** Radio/X-ray correlation for the 'outlier' track of H1743-322 (in the upper x-axis, the inner-disk luminosity is $L_{\rm in}$). The black circles represent the observed radio/X-ray correlation, while the blue squares represent the radio/inner-disk correlation where the inner-disk luminosity is calculated using Equations (4) and (5).

### 3.2. MAXI 1348-630

MAXI J1348-630 was discovered as a bright X-ray transient on 26 January 2019 [50] by the MAXI monitor onboard the International Space Station [51]. Thanks to immediate and intense multiwavelength follow-up observations, it was subsequently identified as a BH candidate [52,53]. A source distance of $D = 2.2^{+0.5}_{-0.6}$ kpc was measured from observations of H I absorption [54]; with this distance, the BH mass can be estimated as $M_{\rm BH} \approx 16(D/5{\rm kpc})M_{\odot} = 7M_{\odot}$ [55]. Recently, Jia et al. (2022) [48] reported that the spin parameter is $a_* = 0.78^{+0.04}_{-0.04}$ and that the inclination angle of the inner disk is $i = 29.2^{+0.3}_{-0.5}$. MAXI J1348-630 is, after H1743-322, only the second source in the steep ($L_{\rm R} \propto L_{\rm X}{}^{\mu}$ with $\mu \gtrsim 1$) regime to have such detailed monitoring, allowing its radio/X-ray behavior to be well constrained.

Unlike H1743-322, the radio/X-ray plane of MAXI J1348-630 shows a relatively smooth curve, and it is difficult to determine where the transition luminosity is. Thus, we plot the photon index $\Gamma$ and $L_{\rm X}$ distribution of MAXI J1348-630 here in order to seek the transition regime. Figure 4 shows the relationship between the photon index and the X-ray luminosity of MAXI J1348-630. The values of $\Gamma$ and the X-ray luminosity were obtained from Carotenuto et al. (2021) [32]. We only plot the data points for the hard state (HS) and intermediate state (IMS) in which $\Gamma$ is available and not fixed during the fitting. We find that $\Gamma$ is positively and negatively correlated with $L_{\rm X}/L_{\rm Edd}$ when $L_{\rm X}/L_{\rm Edd} \gtrsim 10^{-3}$ and $L_{\rm X}/L_{\rm Edd} \lesssim 10^{-3}$, respectively, which implies that there is a phase transition around $10^{-3}L_{\rm Edd}$. This result is consistent with Yang et al. (2015) [56], who obtained their results from a large sample of BHXBs and AGNs. Carotenuto et al. (2021) [11] reported that the transition luminosity $L_{\rm tran} \sim 6.3 \times 10^{35}$ erg s$^{-1}$ (nearly $10^{-3}L_{\rm Edd}$), thus, we set the transition luminosity to $10^{-3}L_{\rm Edd}$, i.e., $L_{\rm tran} = 8.8 \times 10^{35}$ erg s$^{-1}$.

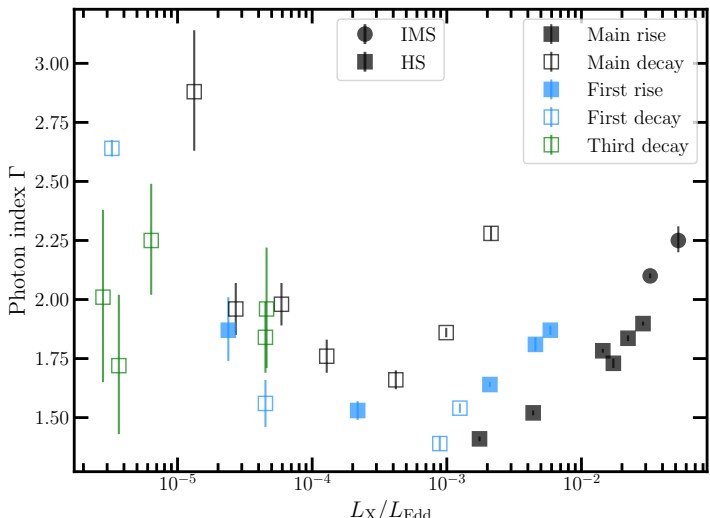

**Figure 4.** The photon index $\Gamma$ versus the 1–10 keV X-ray luminosity (in Eddington units) of MAXI J1348-630. The circles represent the intermediate state (IMS) and the squares represent the hard state (HS). The filled and open symbols indicate separate outburst states, i.e., rise and decay, while different colors differentiate between different outbursts.

We show the distribution of the $L_{in}/L_{bol}$ versus the $L_X/L_{Edd}$ for MAXI J1348-630 in Figure 5. For this source, we have no bolometric luminosity for MAXI J1348-630. The bolometric luminosity in BHXBs can be scaled to the bolometric X-ray luminosity, as the multi-wavelength spectra of all known BHXRBs are dominated by the X-ray band of $0.1$–$2 \times 10^3$ keV [43]. In low-luminosity active galactic nuclei (LLAGNs), the bolometric disk luminosity is estimated from the hard X-ray luminosity, which is believed to be produced by the nuclear region ($L_{bol} \approx 16 L_X$ in the narrow band of 2–10 keV [57]). The $L_{bol}$ can be estimated from the X-ray bolometric correction factor, $f_X = L_{bol}/L_X$, depending on the broad-band spectrum energy distribution (SED). Here, we use $f_X = 5$ [58] for the 'outlier' track of MAXI J1348-630. Several results can be derived directly from this plot. First, the relationship between $L_{in}/L_{bol}$ and $L_X/L_{Edd}$ for MAXI J1348-630 is different from that of H1743-322. There is a tendency of $L_{in}/L_{bol}$ to decrease with $L_X/L_{Edd}$ in the faint end, while $L_{in}/L_{bol} \sim 0.05$ is constant with a more luminous $L_X/L_{Edd}$. On the other hand, $L_{in}/L_{bol}$ varies between $\sim 0.05$ and $\sim 0.25$, with a mean value of $<L_{in}/L_{bol}> = 0.111 \pm 0.005$, which is much lower than that in H1743-322. Furthermore, $L_{in}$ is one magnitude lower than $L_X/L_{Edd}$. The high spin parameter, $a_* = 0.78$, contributes to the low $L_{in}$ from Equation (4). This may suggest that the inner disk is extremely small and contributes only slightly to X-ray luminosity.

Figure 6 shows the 'outlier' track of MAXI J1348-630. The symbols have the same meaning as in Figure 3. Although the inner disk–jet coupling shows good consistency with the radio/X-ray correlation, $L_{in}$ is about one order of magnitude fainter than $L_X$. Based on Equation (4), $P_{jet} \propto a_*^2 L_{in}$, as the radiative efficiency $\varepsilon$ is nearly a constant during the high accretion regime, which means that a higher $a_*$ with the same $P_{jet}$ will lead to the a lower $L_{in}$. The $L_R$ and $L_X$ of MAXI J1348-630 are similar to those of H1743-322 during their 'outlier' tracks, while the spin parameter of MAXI J1348-630 is four times that of H1374-322. From this point of view, we can suggest that the inner-disk of MAXI J1348-630 is less luminous, probably due to its small physical size, which is consistent with our assumption for Equation (4). In addition, it is possible that MAXI J1348-630 has a relatively weak magnetic flux $\Phi/\Phi_{MAD} \lesssim 0.25$, i.e., it is not in the MAD state.

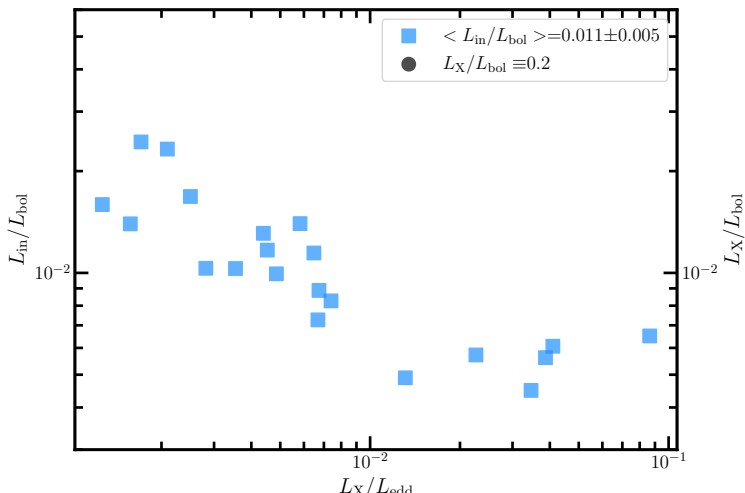

**Figure 5.** Distribution of the inner-disk luminosity to bolometric luminosity ratio $L_{in}/L_{bol}$ versus 1–10 keV X-ray luminosity (in Eddington units) in the 'outlier' track of MAXI J1348-630 (in the right y-axis, the 1–10 keV X-ray luminosity to bolometric luminosity ratio is $L_X/L_{bol}$). As labelled in the plot, the blue squares represent $L_{in}/L_{bol}$; $L_X/L_{bol}$ is set to 0.2, i.e., the X-ray bolometric correction factor $f_X = 5$ [58].

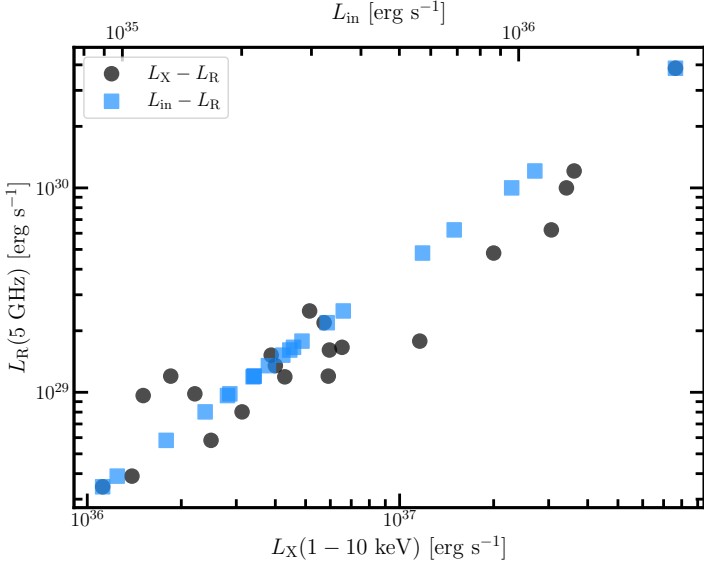

**Figure 6.** Radio/X-ray correlation for the 'outlier' track of MAXI J1348-630 (in the upper x-axis, the inner-disk luminosity is $L_{in}$). The black circles represent the observed radio/X-ray correlation, while the blue squares represent the radio/inner-disk correlation.

## 4. Discussion

### 4.1. Jet Power and Jet Efficiency

From Equation (2), we note that the BZ-jet power is proportional to the square of the BH spin parameter only if the other parameters are constants or the same for different BHXBs. In the MAD state, the only other parameter, except for the BH spin, is the mass accretion rate; thus, for similar accretion rates, e.g., at a high accretion regime of outbursts, the BZ-jet power could be roughly proportional to the square of the BH spin parameter. However, the mass accretion rate $\dot{M}_{BH}c^2$ in the inner accretion disk is difficult to measure; instead, we use $L_{in} \approx \varepsilon\dot{M}_{BH}c^2$, as shown in Equation (4). We are then able to constrain the inner-disk luminosity $L_{in}$ (presented by the coronal X-ray luminosity), assuming that the hot accretion flow (HAF) reached the MAD state, meaning that the $\varepsilon$ of the HAF [37] can be used in the equation. Additionally, jet power has been defined differently, which may

have led to uncertainty in the $P_{\text{jet}} - a_*$ correlation, such as the peak luminosity used by Narayan and McClintock (2012) [24] and Steiner et al. (2013) [25], the total flare intensity used by Russell et al. (2013) [26], and the formula used by Heinz and Grimm (2005) [42], which we use in this paper. Furthermore, in addition to the jet power from the innermost accretion regime around a spinning BH such as the BZ-jet, the observed jet power might partly originate from the accretion disk at larger scales, e.g., in the simulations of standard and normal evolution (SANE) with non-spinning BH [24] and a purely radiative pressure-induced jet [59]. This fraction of jet power would be difficult to distinguish from the BZ-jet power in Equation (2), causing uncertainty in the $P_{\text{jet}} - a_*$ correlation.

The kinetic jet power of BHXBs, which was formally derived by Heinz and Grimm (2005) [42] and which we used in our study, i.e., $P_{\text{jet}} \propto L_{\text{R}}^{12/17}$, can lead to the $L_{\text{R}} \propto (L_{\text{X}}^{\text{in}})^{\sim 1.4}$ in Equation (4) for $L_{\text{X}}^{\text{in}} \approx L_{\text{in}}$ and a nearly constant $\varepsilon \approx 0.08$ in the high accretion regime [37]. The resulting correlations from the two 'outliers' in Figures 3 and 6 seem to both support their MAD state in the flare rising phase while supporting the kinetic power formula of Heinz and Grimm (2005) as well [42].

The jet efficiency, $\eta_{\text{jet}}$, which characterizes the fraction of accretion power that enters the relativistic jet, is an important quantity in accretion theory. It is defined as $\eta_{\text{jet}} = P_{\text{jet}}/(\dot{M}_{\text{BH}}c^2)$. As shown in Section 3, using the radiative efficiency $\varepsilon = 0.08$ together with Equation (3), the jet efficiency can be re-expressed as

$$\eta_{\text{jet}} = \frac{\varepsilon P_{\text{jet}}}{L_{\text{bol}}}, \tag{6}$$

Here, we use the total bolometric luminosity $L_{\text{bol}}$ instead of the inner-disk luminosity $L_{\text{in}}$, as $L_{\text{in}}$ is a function of the jet power $P_{\text{jet}}$ from Equation (4).

Figure 7 shows the plot of the relationship between jet efficiency, $\eta_{\text{jet}}$, and $L_{\text{X}}/L_{\text{Edd}}$ for both sources in their 'outlier' track; the blue circles represent H1743-322 and the orange squares represent MAXI J1348-630, as labeled in the plot. Here, we estimate the bolometric luminosity $L_{\text{bol}}$ from the X-ray bolometric correction factor $f_{\text{X}} = 5$, as in Section 3. First, those two sources show different tendencies; H1743-322 shows an uptrend correlation, while the $\eta_{\text{jet}}$ of MAXI J1348-630 is negatively correlated with $L_{\text{X}}/L_{\text{Edd}}$ when $L_{\text{X}}/L_{\text{Edd}} \lesssim 10^{-2}$ and is not correlated with $L_{\text{X}}/L_{\text{Edd}}$ when $L_{\text{X}}/L_{\text{Edd}} \gtrsim 10^{-2}$. The mean jet efficiency of H1743-322 and MAXI J1348-630 is, respectively, $0.011 \pm 0.005$ and $0.007 \pm 0.003$. The $\eta_{\text{jet}}$ of MAXI J1348-630 shares a similar tendency to the result of Xie and Yuan (2016) [10], which were obtained from the LHAF (luminous hot accretion flow)–jet model. However, our result is one magnitude lower than that of Xie and Yuan (2016) [10]. Obviously, the jet efficiency depends on the magnetic flux and the BH spin [14,15], as the jet power has a close connection to both parameters. Our results for the mean $\eta_{\text{jet}}$ of H1743-322 are slightly shallower than the results reported in Narayan et al. (2022) [15]. They performed nine numerical simulations of MADs across different values of the black hole spin parameter and found that the $\eta_{\text{jet}}$ of the BZ-jet is negatively and positively correlated with black hole spin when $a_* < 0$ and $a_* > 0$, and the $\eta_{\text{jet}} \sim 0.03$ when $a_* = 0.2$. We would note that the $\eta_{\text{jet}}$ of Narayan et al. (2022) [15] was measured from $5R_{\text{g}}$ and that they used the total outflowing power $P_{\text{out}}$ instead of the power in a relativistic jet $P_{\text{jet}}$. Here, we obtain the $\eta_{\text{jet}} \sim 0.022$ of H1743-322 by assuming a total accretion disk radius of $\sim 20R_{\text{g}}$ for our $L_{\text{bol}}$ with $\dot{M}_{\text{BH}} \propto r^s$, where $s \sim 0.4$–$0.5$ [38]. This result may suggest that H1734-322 is in the MAD state. However, for MAXI J1348-630, the mean $\eta_{\text{jet}}$ is two orders of magnitude lower than the simulation result in Narayan et al. (2022) [15], $\eta_{\text{jet}} \sim 0.9$. This result suggests that the MAD state may not exist in MAXI J1348-630, even when considering the additional outflowing power and the larger radius of the accretion flow.

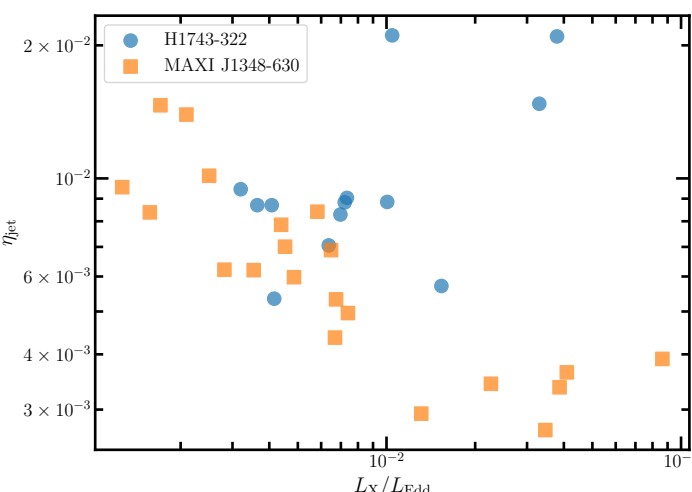

**Figure 7.** Distribution of the jet efficiency versus the 1–10 keV X-ray Eddington ratio $L_X/L_{Edd}$ in the 'outlier' track. The blue circles represent H1743-322 and orange squares represent MAXI J1348-630.

### 4.2. MAD in BHXBs

It is generally believed that the standard track in BHXBs is caused by the hot accretion flow (e.g., [10,12,60]). As shown in Section 3, the typical 'outliers' track is possibly caused by the BZ-jet at approximately the MAD state. The BZ-jet and inner-disk coupling show a good fit to H1743-322 and MAXI J1348-630. One notable limitation of this work is that we aggressively omit the magnetic flux of both sources and assume them under the MAD state, as we have no measurements of the magnetic field from either source. Additionally, an important source of uncertainty in this picture is that the inner-disk luminosities of the two BHXBs are quite different, and it remains unclear whether or not both sources have a high magnetic flux. Furthermore, the MAD state might not be maintained in the decaying phase of a flare. The radio emission at a lower accretion regime thereby tends to rejoin the standard track, as shown in Figure 1, which might be partly caused by the hot accretion flow.

Furthermore, if a BHXB has not reached the MAD state, a jet might be produced from a sub-MAD system or by the accretion disk itself at larger scales, e.g., in a purely radiative pressure-induced jet [59]. The standard track with $L_R \propto L_X^{0.6}$ could originate from the non-MAD state or a sub-MAD state, e.g., in GX339-4, which showed mainly a standard track (though an 'outliers' component has been posited by Islam and Zdziarski (2018) [43]). A sub-MAD state could exist between the MAD and SANE states, with a magnetic flux threading on the BH that is not being saturated and the magnetic field not strong enough, leaving the inner accretion disk not fully magnetically arrested. Liu et al. (2016) [61] assumed the magnetic flux of a sub-MAD state as a power-law function of that in the MAD state. If we assume $\Phi \approx \Phi_{MAD}^\kappa$ and $\kappa \lesssim 1$ for a magnetic flux $\Phi \lesssim \Phi_{MAD}$ in a sub-MAD state and consider $L_{Edd} = \varepsilon_{Edd}\dot{M}_{Edd}c^2 \approx 1.38 \times 10^{38}(M_{BH}/M_\odot)$ erg s$^{-1}$, where the Eddington radiative efficiency $\varepsilon_{Edd} \approx 0.1$ [62], $L_{in}$ can be approximately $L_X^{in}$. Together with Equation (1), we then have

$$P_{jet} \propto a_*^2(1 + 0.85a_*^2)(M_{BH}/M_\odot)^{2(\kappa-1)}(L_X^{in}/\varepsilon)^\kappa \tag{7}$$

In this formula, for a spinning BH in a sub-MAD state, the jet power may have a flatter power-law index with an inner X-ray luminosity (if the disk radiative efficiency $\varepsilon$ for the sub-MAD and the MAD state is similar) as compared to Equation (4) in the MAD state. For the relation $L_R \approx 6.1 \times 10^{-23}P_{jet}^{17/12}$ erg s$^{-1}$ with Equation (7), we have $L_R \propto (L_X^{in}/\varepsilon)^{1.4\kappa}$. The standard track with $L_R \approx L_X^{\sim 0.6}$ can then be explained if $\kappa$ is significantly less than unity. We note that in Equation (7) the jet power is inversely correlated with the BH mass if $\kappa < 1$; however, for an individual BHXB, its BH mass is a constant, and it would be much more effective for different supermassive BHs in AGNs. More studies are required for

a sub-MAD system with a spinning BH, regardless of whether or not our assumption is reasonable.

**5. Summary and Outlook**

In this work, we focus on the 'outlier' tracks of radio/X-ray correlation in H1743-322 and MAXI J1348-630. Both sources share similar radio and X-ray luminosity during their 'outlier' tracks. We assume the MAD state with the inner accretion disk in Equation (2) to calculate the inner-disk luminosity and try to explain the 'outlier' track with the BZ-jet in the MAD state.

Our main results can be summarized as follows:

- The BZ-jet in the MAD state and the inner-disk coupling show good consistency with the observed radio/X-ray correlation in both sources. This suggests that the BZ-jet might explain the 'outlier' tracks of both sources. While the accretion disk of H1743-322 could be in the MAD state, there is a lower possibility that MAD is achieved in MAXI J1348-630 due to its low jet production efficiency.
- There is a phase transition regime around $L_X/L_{Edd} \sim 10^{-3}$ of MAXI J1348-630, as shown in Figure 4. This result is consistent with Yang et al. (2015) [56] and Carotenuto et al. (2021) [11], suggesting a transition luminosity $L_{tran} \approx 8.8 \times 10^{35} \, \mathrm{erg \, s^{-1}}$.
- The bolometric luminosity ratios $L_{in}/L_{bol}$ of H1743-322 and MAXI J1348-630, i.e., $0.191 \pm 0.081$ and $0.011 \pm 0.005$, respectively, are quite different, implying that the latter is in a relatively low state.

We would note that although the result of this work shows that a BZ-jet in the MAD state with inner-disk coupling could account for the 'outlier' track in BHXBs, it is not a strictly numerical simulation of the MAD state. Thus, general relativistic magneto-hydrodynamic (GRMHD) simulations (e.g., [15,36]) are needed to explore the accretion of the BHXBs. Future observations and analyses should focus on the magnetic field in BHXBs, the bolometric luminosity, and, if possible, the physical properties of the inner disk (e.g., temperature and physical size).

**Author Contributions:** Conceptualization, N.C. and X.L.; methodology, N.C. and X.L.; formal analysis, N.C., X.L. and F.-G.X.; writing—original draft preparation, N.C. and X.L.; writing—review and editing, N.C., X.L., F.-G.X., L.C. and H.S.; visualization, N.C. All authors have read and agreed to the published version of the manuscript.

**Funding:** This work was supported by the National Key R&D Program of China under grant number 2018YFA0404602. FGX was supported in part by the National SKA Program of China (No. 2020SKA0110102) and the Youth Innovation Promotion Association of CAS (Y202064). L.C. was supported by the Chinese Academy of Sciences (CAS) 'Light of West China' Program, under grant No. 2021-XBQNXZ-005. H.S. was supported by National Natural Science Foundation of China under grant number 11673056 and 11173042.

**Institutional Review Board Statement:** Not applicable.

**Data Availability Statement:** Not applicable.

**Acknowledgments:** We appreciate the referees for their helpful comments. This paper is dedicated to the memory of Tan Lu, who supervised Xiang Liu from September 1994 to July 1996 at Nanjing University.

**Conflicts of Interest:** The authors declare no conflict of interest.

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
