# Peer review of "Explaining the ‘Outliers’ Track in Black Hole X-ray Binaries with a BZ-Jet and Inner-Disk Coupling"

_universe, doi:10.3390/universe8060333_

Round 1

Reviewer 1 Report

This essay deals with the accretion-ejection relationship of two stellar-mass black holes, explained by means of the Blandford-Znajek and magnetically-arrested disk theories. The essay is well written, the starting hypotheses and assumptions are explicitly declared, the conclusions are consistent with the above cited point. I have no objection for the publication on Universe journal in the present form.

Author Response

Dear Referee,

This essay deals with the accretion-ejection relationship of two stellar-mass black holes, explained by means of the Blandford-Znajek and magnetically-arrested disk theories. The essay is well written, the starting hypotheses and assumptions are explicitly declared, the conclusions are consistent with the above cited point. I have no objection for the publication on Universe journal in the present form.

Re:  Thanks a lot, we have included two references, and improved our English presentation.

Reviewer 2 Report

It will be worthful to recall  the two old (possible first) papers by (soviet) astrophysics Bisnovatyi-Kogan and Ruzmaikin ( ApSS, 1974,1976) about a  accretion flow model.

Author Response

Dear Referee,

It will be worthful to recall  the two old (possible first) papers by (soviet) astrophysics Bisnovatyi-Kogan and Ruzmaikin ( ApSS, 1974,1976) about a  accretion flow model.

Re:  Thanks a lot, we have included the two references, and also improved our English presentation.

Reviewer 3 Report

The authors analyze the astrophysical properties of the so-called outlier black hole x-ray binary systems and try to explain their behavior as a consequence of magnetically arrested disks. The explanation works well in one of the systems, H1743-322, but not in the other, MAXI J1348-630. Still, I find it an interesting and honest attempt with the potential for a more refined analysis in the future. Therefore, I would recommend the manuscript for publication. However, the presentation needs thorough proofreading, including, but not limited to, spell-check and grammar-check, to improve readability. 

Author Response

Dear Referee,

The authors analyze the astrophysical properties of the so-called outlier black hole x-ray binary systems and try to explain their behavior as a consequence of magnetically arrested disks. The explanation works well in one of the systems, H1743-322, but not in the other, MAXI J1348-630. Still, I find it an interesting and honest attempt with the potential for a more refined analysis in the future. Therefore, I would recommend the manuscript for publication. However, the presentation needs thorough proofreading, including, but not limited to, spell-check and grammar-check, to improve readability. 

Re: Thanks indeed, we have included two references, and the Journal Editor helped improved our English presentation. Sorry, the changes of English are small and not marked in bold face.